# Traumatic Events and Eagle Syndrome: Is There Any Correlation? A Systematic Review

**DOI:** 10.3390/healthcare9070825

**Published:** 2021-06-29

**Authors:** Sabina Saccomanno, Vincenzo Quinzi, Nicola D’Andrea, Arianna Albani, Licia Coceani Paskay, Giuseppe Marzo

**Affiliations:** 1Department of Life Health and Environmental Sciences, University of L’Aquila, 6700 L’Aquila, Italy; vincenzo.quinzi@univaq.it (V.Q.); nicola.dandrea@student.univaq.it (N.D.); arianna.albani@student.univaq.it (A.A.); giuseppe.marzo@univaq.it (G.M.); 2Academy of Orofacial Myofunctional Therapy, Pacific Palisades, CA 90272, USA

**Keywords:** Eagle syndrome, traumatic events, styloid process

## Abstract

Background: Eagle syndrome occurs when elongated styloid process or ossification of the stylohyoid ligament interfere with the surrounding anatomical structures giving rise to various symptoms. Watt W. Eagle identified two types: stylo-hyoid classic syndrome and stylo-carotid artery syndrome. The aim of this systematic review of the literature is to evaluate correlations between Eagle syndrome and traumatic events or teeth extractions. Methods: out of 294 articles, the final study allowed the identification of 13 studies focusing on traumatic events. Out of 342 articles, the final study allowed the analysis of two studies regarding extractive dental events. Results: 13 articles showed correlations between the onset of symptoms in Eagle syndrome and traumatic events and highlighted two possibilities: traumatic event could fracture the already elongated styloid process or calcified stylohyoid ligament; trauma itself triggers the pathophysiological mechanisms that lead to lengthening of styloid process or calcification of stylohyoid ligament and therefore the typical symptoms. The only two case reports concerning Eagle syndrome symptoms after extractive dental events describe the onset of classic type. Conclusions: The analyzed articles confirm correlation between traumatic event and onset of typical symptoms of Eagle syndrome. There is not enough literature linking extractive dental events to Eagle syndrome. Trial registration: CRD42020185176.

## 1. Introduction

Oro-facial pain is among the most common manifestations of pain reported by patients. In the past, little attention was given to the facial pain and headache resulting from an elongated styloid process [1].

Eagle syndrome (ES) refers to a rare constellation of symptoms caused by an abnormally long styloid process or stylohyoid chain ossification, which is characterized by craniofacial or cervical pain. It was first described by an Italian surgeon, Pietro Marchetti, in 1652; however, the clinical syndrome was definitively outlined by Watt Weems Eagle in 1937 [2].

An elongated process is present in approximately 4% of the population, and the vast majority of these are asymptomatic. Eagle estimated the prevalence of symptoms in patients with elongated styloid processes as 0.16%. The syndrome has a female-to-male predominance of three to one. It generally occurs in adult patients, with the age ranging from 30 to 50 years [3,4]. The styloid process elongation is often bilateral. However, when symptomatic, the symptoms are mostly unilateral.

Eagle described two types of the syndrome:
-*“Stylo-hyoid classic syndrome”*, which is characterized by pharyngeal pain aggravated by swallowing and frequently referred to the ear [2]. Other typical symptoms are headache and the sensation of a foreign body in the throat. Symptoms were thought to be due to impingement of cranial nerves (trigeminal, facial, glossopharyngeal or vagus) by the tip of the styloid process [5].-*“Stylo-carotid artery syndrome”* in which its vascular form is attributed to impingement of the internal carotid artery, extracranially, by the styloid process. This can cause a compression when turning the head or in “dissection” of the carotid artery resulting in a transient ischemic accident or stroke.


### 1.1. Etiopathogenesis

In order to recognize and then treat this pathological condition, it is necessary to know the anatomy and embryology of the styloid process and the dynamics that come into play when a traumatic event occurs.

Anatomically, the styloid process is a thin, elongated, cylindrical bone projection in an anteromedial position with respect to the mastoid process of the temporal bone. Its length varies between 20 and 30 mm and connects to three muscles and two ligaments. The muscles that originate from it are the styloglossus, the stylohyoid and the stylopharyngeal, which extend to the tongue, the hyoid bone and the pharynx respectively. The styloglossum and stylohyoid are innervated by the hypoglossal and facial nerves while the stylopharyngeal is innervated by the glossopharyngeal nerve. The ligaments that attach to the styloid process are the stylohyoid and the stylomandibular. The stylohyoid ligament originates from the end of the styloid process and extends to the small horn of the hyoid bone. The stylomandibular ligament extends from the styloid process to the posterior edge of the mandible; both are helpful in the movements of the mandible, the hyoid bone, the tongue and the pharynx.

Embryologically, the styloid process, stylohyoid ligament, lesser cornu of the hyoid bone and the superior portion of the hyoid body are derived from Reichert’s cartilage, which arises from the second pharyngeal arch. The styloid process begins to ossify at the end of pregnancy and continues to undergo calcification over the first 8 years of life. [2]

Several theories were proposed by Steinmann between 1968 and 1970 to explain etiopathogenesis, which are considered valid even today:
-Theory of Reactive Hyperplasia.-Theory of Reactive Metaplasia.-Theory of Anatomic Variance [2].


In some cases, Eagle syndrome symptoms occur without an identifiable etiology or they could be consequences of tonsillectomy [6]. The observation of symptoms after tonsillectomy generates the hypothesis that cranial nerves V, VII, IX or X are entrapped in the locally formed granular tissue. Trauma to the soft tissues during tonsillectomy may cause bone formation, leading to an elongated styloid process or ossified stylohyoid ligament. Ossification typically appears from 2 to 12 months after the trauma [2].

### 1.2. Diagnosis

Since the symptoms are variable and nonspecific, patients usually seek treatment from several different specialties such as otolaryngology, maxillofacial surgery, neurology, neurosurgery, and finally psychiatry [7].

The diagnosis of Eagle’s syndrome is based on four different parameters:
Clinical manifestations.Digital palpation of the process in the tonsillar fossa.Radiological findings.Lidocaine infiltration test.


In examining the clinical manifestations it is difficult to reach the correct diagnosis only on the basis of symptoms referred by the patients. It is important for physicians and dentists to consider Eagle’s syndrome in the differential diagnosis, since symptoms such as headache, facial pain, and neck pain, overlap with other clinical conditions such as diseases of nose, ears and throat and other pathologies, ranging from psychosomatic diseases to neoplasms.

When carrying out the objective examination we mainly use tonsillar fossa palpation and Eagle himself considered this test as a diagnostic indicator. Digital palpation, in fact, immediately reproduced pain referred by the patients. However, no controlled study has proven the sensitivity or specificity of this indicator. The styloid process can only be palpated in the tonsillar fossa if it is longer than 7.5 cm [2].

Confirmation of Eagle syndrome is always radiographic:
-Lateral teleradiography of the head: a disadvantage is that the styloid processes of the two sides may overlap.-Anterolateral modified Towne’s radiograph: this procedure calculates both the medial and lateral deviation of the process.-Panoramic radiograph: using this procedure clinicians can typically consider the styloid process to be elongated if its length is more than 1/3 of the length of the ramus of the mandible. An advantage of the orthopantomograph is that the entire length of the styloid process is visible and its deviation can be measured quite accurately.-Computed tomography: is an effective method for evaluating styloid process length, angulation and other morphological characteristics.-Sagittal computed tomography angiography: in cases of vascular compression, it can also be effective in assessing blood flow disturbance [2].


Saccomanno et al. evaluated different radiographic methods to investigate Eagle syndrome. Validity of panoramic radiograph as a diagnostic method for elongated styloid process and Eagle syndrome was investigated in a systematic review. Panoramic radiograph is a routine exam that gives an overview of the general oral and maxillofacial conditions. It is easy to perform and to interpret, with the advantage of having lower biologic, i.e., radiant dose and economic costs than computed tomography or cone beam computed tomography. For these reasons, it is more appropriate for the first diagnosis and for epidemiological evaluations. Therefore, in symptomatic patients, panoramic radiograph could be of help in the differential diagnosis of other conditions associated with orofacial and neck pain. In fact, the presence of radiographic evidence of styloid process longer than 30 mm, in patients showing pain during mouth opening, deglutition, or head rotation can be suggestive of Eagle’s syndrome [8].

Although plain skull radiographs might be sufficient to reveal the anatomical abnormality, CT of the head/neck and especially 3D-computed tomography scans are considered the gold standard for visualization of the anatomically complex styloid process, as it avoids the problems of obscured overlapping anatomy. Moreover, it underlines the styloid process angulation, which is crucial for the surrounding anatomical relationships [9].

Finally, an additional test consisting of *lidocaine infiltration* can be used (1 mL 2%) in the tonsillar fossa: if the patient’s symptoms and local sensitivity subside, then the test result is considered positive for a diagnosis of Eagle’s syndrome [2].

### 1.3. Objectives

The aim of this work is to investigate literature with a systematic review that evaluates correlation between Eagle syndrome onset and traumatic events. A further aspect that will be examined is the possibility of symptomatic cases of Eagle syndrome following extractive therapies in dentistry. Our work considered only patients who have experienced traumatic events or extractive dental events before Eagle syndrome onset.

## 2. Systematic Review

### 2.1. Protocol and Registration

This systematic review was conducted following the guidelines of the Cochrane Handbook for Systematic Reviews of Interventions [10]. The methods of analysis and the inclusion criteria were specified in advance and documented in a protocol registered in the National Institutes of Health Research Database (http://www.crd.york.ac.uk/prospero/; trial registration number: CRD42020185166).

### 2.2. Eligibility Criteria

The studies included in the present systematic review were: bibliographic reviews, systematic reviews, meta-analysis and case reports analyzing correlation of Eagle syndrome with traumatic events and extractive dental events. We limited the inclusion criteria solely to English-language publications. Articles with unavailable full-text and articles in which Eagle syndrome and traumatic events were uncorrelated articles were excluded as well.

### 2.3. Information Sources, Search Strategy, and Study Selection

This systematic literature review was carried out by searching articles on: PubMed, Cochrane Library, Campbell Collaboration, Current Contents, Turning Research Into Practice (TRIP) and SciELO. The following key words have been used: “Eagle Syndrome”, “Eagle’s Syndrome”, “Elongated Styloid Process”, “Trauma”, “Dental Procedure”, “Extraction” and “Dentistry”. Eligibility was discussed by authors by screening the title and abstracts of the retrieved articles. Whenever in doubt about the inclusion or exclusion of an abstract, the full text was accessed. The search strategies are reported in Table 1.

### 2.4. Data Items

The main outcome of this systematic review is that some cases of Eagle syndrome arise after traumatic event and after tooth removal. Patients can be treated in different ways depending on the type and severity of the syndrome with complete remission of symptoms. We collected the data we found on the articles we finally included.

### 2.5. Summary Measures and Approach to Synthesis

We included review articles, systematic reviews, meta-analysis articles and case reports in which patients showed Eagle syndrome typical symptoms after traumatic events and tooth removal as well. After full-text reading we extracted data concerning sex, age, type of traumatic event, symptoms, diagnostic devices, final diagnosis and treatment of the patient. After data extraction we recorded all of them in three tables. (Results, Table 2, Table 3 and Table 4).

Reviewing different search engines resulted in 294 articles available for consideration. After eliminating duplicate articles, 285 articles remained to be screened. After a first reading we further excluded 253 articles because they were not relevant and we evaluated 32 full-text articles. After evaluation of these 32 full-text articles, we excluded 18 articles because trauma was mentioned among the possible causes of Eagle syndrome, but in the specific reported cases it was not correlated to the patients’ symptoms. We also excluded the article by Balbuena et al. (1997) because it was already mentioned in the 2009 review by Piagkou et al. [5]. We finally included 13 articles published between 1978 and 2018 (Figure 1a).

Regarding the trauma of tooth extraction, reviewing different search engines resulted in 342 articles available for consideration. After eliminating duplicate articles, 340 articles remained to be screened. After a first reading we excluded 319 articles because they were not relevant and we evaluated 21 full-text articles. After evaluating these 21 full-text articles, we excluded 19 articles because they did not show any correlation between Eagle syndrome and extractive dental events. This left two articles published between 2016 and 2018 in which extractive dental events seemed to be correlated to Eagle syndrome (Figure 1b).

### 2.6. Risk of Bias

The studies were graded under high, uncertain or low risk, based on: Selection bias, Performance bias, Detection bias, Attrition bias and Reporting bias. The quality of individual studies was evaluated based on the categorized ranking by Oxford Centre for Evidence-Based Medicine 2011 Levels of Evidence.

## 3. Results

### 3.1. Correlation between Eagle Syndrome and Traumatic Events

Our systematic review of the literature highlighted 15 articles showing correlations between the onset of symptoms of Eagle syndrome and traumatic events.

These articles date from 1978 to 2018 and are mainly case reports [1,3,5,6,7,11,12,13,14,15,16] a review [2] and case report and systematic review article [17].

In the case reports included in this systematic review, several traumatic events such as car or motorbike accidents, aggressions, blunt traumas or accidental falls are mentioned and the most commonly found syndromic form is the classic type characterized by symptoms such as headache, dysphagia, cervical pain, pharyngeal pain and foreign body sensation [1,3,6,7,14,15,16,17]. In 2018 Pèus et al. reported a case of typical vascular symptoms in a patient with neurological damages [11]. The three remaining case reports, instead, highlight vascular consequences in Eagle syndrome that can lead to carotid artery “dissection” and ischemic, palsy and stroke conditions [5,12,13]. Piagkou et al. in their 2009 systematic review evaluated physiopathological mechanisms underlying the lengthening of the styloid process and the ossification of the styloid ligament [2].

The diagnosis of Eagle syndrome is based on anamnesis, objective examination and instrumental examination. Through the collection of anamnestic data from patients, it is possible to retrace the traumatic events that led to the manifestation of the pathology: blunt traumas in four articles [1,6,11,17], road accidents (5.2) [13,14], “crack” while chewing [16], neck manipulation [12], aggressions (5.3) [3,16], accidental falls (5.1) [7].

While performing objective examination, in one case it was possible to observe directly the presence of a mass or a bone protrusion due to the elongated styloid process by means of oral observation alone [16]. In other cases it was the palpation of the tonsillar fossa that reproduced the discomfort complained by the patients observed in the studies [1,6,15,16].

In all the articles reviewed, Eagle syndrome was confirmed thanks to instrumental examination: panoramic radiograph [7,14,16,17], Panorex radiograph [6], Lateral teleradiography of the head [1,6,14], head and neck computed tomography [3,5,11,13,15,16], cone beam computed tomography [7] and in case of the vascular form of the syndrome, clinicians also made use of magnetic resonance imaging [11,12], magnetic resonance imaging angiography [12], angiocomputed tomography [13] and electroneurography [11].

The findings highlighted by these investigations and the treatment of patients described in the studies included in the systematic review have been summarized in the previous Table 2, Table 3 and Table 4.

### 3.2. Correlation between Eagle Syndrome and Extractive Dental Events

Our systematic review of the literature concerning Eagle syndrome symptoms after teeth extraction highlighted only two case reports relevant to what we researched: an article by Sowmya et al. [4] and one by Li et al. [18].

These two studies have similarities, as both patients described are males under 40 years of age (38 and 36 years of age respectively) who began to show symptoms typical of the classic form of Eagle’s syndrome following teeth extraction.

In the first article, the patient presented for observation with acute pain on the right side of the face which radiated to the temporal and neck ipsilateral area, also causing a decrease in rotational movements of the neck and head in that direction. He also reported stress and insomnia for about three months, a period during which he underwent lower right jaw third molar extraction.

The diagnosis of Eagle syndrome was obtained through:
-Objective examination: extraoral palpation of a bony mass at the level of the tonsillar fossa area.-Instrumental examinations: panoramic radiograph and computed tomography scans of the neck.


From the examination of the panoramic radiograph the styloid processes were found to be bilaterally elongated and ossified while from the computed tomography display it was possible to measure them: right = 48 mm, left = 40 mm. In this case clinicians opted for drug therapy with diazepam for 5 days, NSAIDs and TENS [4].

In the second article, the patient went in for observation five days after the onset of acute algic symptoms, dysphagia, foreign body sensation with repeated emetic episodes. The patient reported that the symptoms started two hours after the extraction of a dental element (however not specified).

The diagnosis of Eagle syndrome was obtained through a computed tomography scan of the neck, which made it possible to examine the bilaterally elongated styloid processes and measure them: right = 46 mm and left = 47 mm. The patient was treated empirically with analgesic drugs and invited to go back for check-ups, to which he did not comply [18]. Results have been summarized in Table 3.

## 4. Discussion

Eagle syndrome was first described in 1652 by the Italian surgeon Pietro Marchetti, but still there is confusion about this pathological condition because of its various symptoms, which are very common to other diseases.

Once the patient with oral-facial pain presents to us, countless scenarios open up that may involve dentists, otolaryngologists, maxillo-facial surgeons, neurologists, neurosurgeons, psychiatrists, sleep disorder specialists and breathing specialists.

The symptoms described by patients suffering from Eagle syndrome, and in particular in its classic form, are non-specific and common to other pathological conditions:

*Headaches*: migraine, cluster headache, cervicogenic and chronic tension headaches, carotidynia, atypical facial pain, paroxysmal migraine, SDB, hormonal imbalance, allergies affecting breathing.

*Facial pain*: disorders of the temporomandibular joint including joint clicks; neuralgia affecting the glossopharyngeal, trigeminal, and superior laryngeal nerves, the pterygopalatine ganglion, the intermediate nerve and genicular ganglion; myofascial pain syndrome; third molar disodontiasis; incongruous prostheses; salivary gland disorders.

*Neck pain*: degenerative disc pathology, chronic laryngo-pharyngeal reflux, compensatory postural positions such as forward neck posture, and poor ergonomics.

*Pathologies of the ear, nose and throat*: chronic tonsillitis, tonsillar stones, spasms of the constrictor muscles of the pharynx, otitis, mastoiditis, fracture of the hyoid bone, bursitis of the pterygoid hook.

*Other pathologies*: psychosomatic disorders, foreign bodies, inflammatory and neoplastic processes in the oropharyngeal area, pharyngeal and tongue base tumors, cervical arthritis, temporal arteritis, cellulite and nuchal fibrosis, head and tongue syndrome, granular cell tumors.

It also happens that patients with the symptoms mentioned above who have been diagnosed with Eagle syndrome typically have already visited several specialists to try to solve their problems, but without obtaining relief. In the past years, patients affected by Eagle syndrome were not given a specific diagnosis, and the only treatment was pharmacological with analgesics.

Nowadays, however, new 3D diagnostic technologies such as CBCT allow us to precisely visualize and analyze the involved anatomical structures. It is possible, for example, visualize and measure the elongated styloid process or the calcified stylohyoid ligament. It is thanks to advances in the field of radiodiagnostics that there is a greater understanding of pathological conditions such as Eagle’s syndrome that has always been difficult to delineate.

What we want to underline is the importance of differential diagnosis with the above-mentioned diseases and conditions, especially when the symptoms perceived by the patient cannot be attributed to a known cause. It is important, for this reason, that clinicians and dentists consider this syndrome as a possibility when assessing a patient with cervical-facial or cervical-pharyngeal pain [2].

In this systematic review of the literature, we have collected data on the correlation of Eagle’s syndrome, in both its forms, with traumatic events and extractive dental events and we have focused on the methods used to formulate the diagnosis and the subsequent chosen treatment.

What we try to explain through the analysis of the articles is the pathogenetic mechanism by which, following a traumatic event, the styloid process can begin its remodeling giving rise to painful symptoms or how, following a blunt blow or an accidental fall, the styloid process already longer than normal (>30 mm) can fracture and trigger the symptoms.

After the analysis of the X-rays, it is possible to identify a first differentiating element, i.e., the presence of a fracture at the level of the styloid process or ossified styloid ligament. In the articles examined there are seven cases (5.2), [1,6,11,14,15,17] in which fractures of such elongated portions are reported. In these cases the symptomatology is attributable to two distinct mechanisms:
(1)The traumatic fracture of the styloid process causes the proliferation of granulation tissue, which can compress adjacent structures and give rise to typical signs and symptoms.(2)The fractured portion can directly irritate or compress the adjacent noble structures and pharyngeal mucosa.


In the remaining articles considered for this review there was no evident fracture and two distinct scenarios open up, united by trauma, the triggering event. In this case, it is not the fractured portion that causes the symptoms, but the trauma itself that triggers the pathophysiological mechanisms leading to the lengthening of the styloid process or the calcification of the stylohyoid ligament and therefore the typical symptoms. In young patients, on the other hand, these anatomical structures can lengthen without a triggering factor, but they may become additional risk factors in case of traumatic events, thus requiring a differential diagnosis. Therefore we refer back to the hypotheses developed by Steinmann already mentioned in the Introduction:
(1)In cases in which the styloid process is initially normoconforming, the traumatic event may trigger two mechanisms that determine the lengthening of the styloid process and the ossification of the stylohyoid ligament:
Theory of reactive hyperplasia: If the styloid process is properly stimulated by the pharyngeal trauma, ossification can continue from its end towards the stylohyoid ligament. This can occur in the post-trauma healing period and it mainly gives rise to symptoms related to carotid artery impingement.Theory of reactive metaplasia: the traumatic event in this case triggers several metaplastic variations at the level of the stylohyoid ligament that lead to its partial ossification. The eventuality of metaplasia can be attributed to the presence of bone centers, such as bone cells, osteoblasts and osteocytes, between the fibrous formations. In this case the symptoms would originate from the region of the stylohyoid ligament and would affect the soft tissues of the neck giving rise, for example, to dysphagia.
(2)In other cases we can, instead, talk about a Theory of anatomical variability: bone and ligament are qualitatively normal and what varies is the lengthening of the styloid process. This theory explains why it is possible to find early radiographic evidence of this type of ossification in children and young adults who have not suffered any prior cervical-pharyngeal trauma. The fact that these anatomical structures are not normal is a risk factor, as traumatic events may trigger the typical symptoms of Eagle syndrome.


From the systematic review of the literature and the analysis of the described patterns, it is evident that the traumatic event plays a primary role in the pathogenesis of Eagle’s syndrome. The typical symptoms of the syndrome, in fact, can arise following the fracture of already elongated styloid processes or trigger mechanisms that lead to the lengthening or anomalous calcification of the anatomical areas in question that will compress and irritate the surrounding noble structures.

In the specific case of extractive dental trauma there is still little documentation to support the hypothesis that would link them to the onset of Eagle syndrome. There is, in fact, little data available and in one of the two articles reviewed there was no follow-up.

Certainly, the correlation between traumatic events and Eagle syndrome analyzed in our work must lead to consider the elongated styloid process or calcified stylohyoid ligament as risk factors for the onset of this syndrome following an event such as a car accident or prolonged dental treatments.

As this syndrome is often asymptomatic, when such symptoms occur after trauma, the differential diagnosis should include Eagle syndrome in predisposed patients with the above-mentioned anatomical variations.

From the moment the diagnosis of Eagle syndrome is confirmed, the problem should be resolved surgically. However, those who should surgically treat these patients, namely otorhinolaryngologists and maxillo-facial surgeons, are reluctant to do so because there is no specific protocol at this time. Hence, the need for a more precise protocol in the future that takes advantage of the multidisciplinary approach and progress in terms of diagnosis.

A further approach, while waiting for specific surgical guidelines, is that of prevention: using CBCT or panoramic radiographs to identify at-risk patients and to make them aware that trauma could lead to symptoms of Eagle syndrome. Similarly, if these patients are subjected to trauma, it will be less complex to diagnose Eagle’s syndrome and attempt a surgical approach. Further studies could explore this aspect. Our work was, in fact, limited by certain aspects:
-majority of case reports compared to systematic reviews or meta-analyses.-few works in which the syndrome was correlated with traumatic events.-articles correlating Eagle’s syndrome with tonsillectomy [19].


It is clear that the literature on Eagle syndrome has not focused on traumatic events so our aim is to offer insights for further studies.

The last aspect we want to focus on is extractive dental events. There is still little literature about it and we think dentists ought to focus on routine examination such as panoramic radiograph to intercept risk factors for Eagle syndrome. From a forensic point of view, awareness of the risk factors for Eagle syndrome following extractive trauma should be added to the informed consent after prolonged dental sessions or in case of oral surgery.

## 5. Conclusions

This systematic review of the literature makes it clear that Eagle syndrome is a pathology that gives rise to non-specific symptoms that can be traced back to multiple clinical case scenarios and must therefore necessarily be studied and managed by different specialists.

In particular, the dentist can assess the presence of altered anatomical structures such as the elongated styloid process and the calcified styloioid ligament by means of panoramic radiograph or computed tomography scans of the head and neck.

Anamnesis can also help physicians in the differential diagnosis as past trauma may have caused the fracture of the elongated styloid process and the calcified stylohyoid ligament. At the same time the impact can, instead of fracturing a portion, irritate the adjacent soft structures or the pharyngeal mucosa and cause the classic or stylo-carotid syndrome.

The articles found in the literature confirm the strong correlation between traumatic events and the onset of the typical symptoms of Eagle syndrome. As far as the correlation with extractive dental events is concerned, however, there is not enough literature to support a thesis linking this type of dental practice to Eagle syndrome.

Further studies are therefore needed to better explore the correlation between Eagle syndrome and traumatic or dental extractive events and to make clinicians aware of this pathological condition and its implications on patients’ everyday life.

## Figures and Tables

**Figure 1 healthcare-09-00825-f001:**
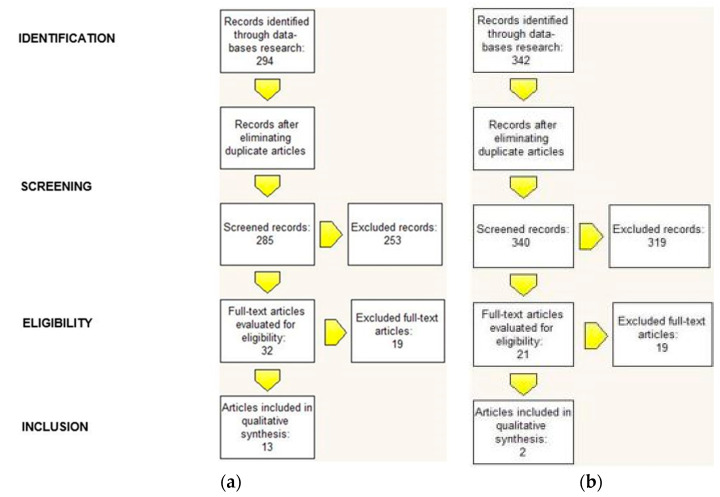
(**a**) Correlation between Eagle Syndrome and traumatic events; (**b**) Correlation between Eagle Syndrome and extractive dental events.

**Table 1 healthcare-09-00825-t001:** Search strategy.

Correlation between Eagle Syndrome and Traumatic Events	Correlation between Eagle Syndrome and Extractive Dental Events
(EAGLE SYNDROME) OR EAGLE’S SYNDROME) OR ELONGATED STYLOID PROCESS) AND (TRAUMA)	(EAGLE SYNDROME) OR EAGLE’S SYNDROME) OR ELONGATED STYLOID PROCESS) AND (DENTAL PROCEDURE) OR TOOTH REMOVAL) OR EXTRACTION) OR DENTISTRY)

**Table 2 healthcare-09-00825-t002:** Summarized correlation between Eagle Syndrome and traumatic events.

Title, Authors [Reference]	Article Type	Year	Sex	Age	Traumatic Event Referred	Symptoms	Diagnostic Tools	Diagnosis	Treatment
Facial pain from an elongated styloid process (Eagle’s syndrome) Massey E.W. [1]	Case Report	1978	F	30	Blunt trauma of left side occipital area	Headache on temporal and occipital area	Lateral Teleradiography of the head: bilaterally elongated styloid processes; left one fractured.	Classic Form	Conservative (no surgical or pharmacological therapy)
Eagle syndrome: an incidental finding in a trauma patient: A case report. Jewett et al. [3]	Case Report	2014	M	~50	Aggression	Neck and back pain; foreign body sensation of right side neck.	CT: elongated right styloid process and ossified right stylohyoid ligament.	Classic Form	Conservative (no surgical or pharmacological therapy)
Eagle Syndrome Presenting after Blunt Trauma, Mann A. et al. [5]	3 Case Reports	2016	(5.1) F (5.2) F (5.3) M	(5.1) 73 (5.2) 39 (5.3) 38	((5.1) Accidental fall (5.2) Motorbike accident (5.3) Aggression	(5.1) dizziness when rotating head to the left (5.2) left hemiparesis (5.3) left side cervical pain; aphasia; foreign body sensation in pharynx.	(5.1) CT: bilaterally ossified stylohyoid ligament; (5.2) CT: fractured right styloid process; ossified stylohyoid ligament; internal carotid dissecation; (5.3) CT: styloid process contacts internal carotid.	Stylo-Carotid Form	(5.1) Unavailable follow-up; (5.2) Pharmacological and conservative therapy; (5.3) Vascular surgery: stent.
A report of post-traumatic Eagle’s Syndrome. Klècha A. et al. [6]	Case Report	2008	F	39	Car accident with whiplash	Right side TMJ pain; pharyngeal pain; right side ear pain; insomnia; cracking sensation	Panoramic Radiograph: bilaterally ossified stylohyoid ligaments.	Classic Form	Surgical intervention to remove styloid process and part of the stylohyoid ligament with transcervical approach.
The Development of Eagle’s Syndrome after Neck Trauma, Shaifulizan AR. et al. [7]	Case Report	2018	F	43	Trauma while chewing	Persistent pain on the right side of the neck irradiated to the face, temporal and occipital area; pain in the right mandibular angle.	Panoramic Radiograph: elongated right styloid process. Cone Beam CT: right styloid process = 48 mm, left styloid processs = 37 mm	Classic Form	Surgical intervention to remove styloid process with transcervical approach.
Recurrent unilateral peripheral facial palsy in a patient with an enlarged styloid process. Peùs D. et al. [11]	Case Report	2018	M	39	Blunt trauma	Episodes of left hemifacial paresis.	Electroneurography: marked comparative amplitude reduction on the left side -> axonal damage of the left facial nerve MRI + CT: elongated and fractured from its basis left styloid process.	Classic Form with Stylo-Carotid Form Symptoms	Surgical intervention to remove styloid process with transcervical approach.

**Table 3 healthcare-09-00825-t003:** Summarized correlation between Eagle Syndrome and traumatic events.

Title	Article Type	Yezar	Sex	Age	Traumatic Event Referred	Symptoms	Diagnostic Tools	Diagnosis	Treatment
*Eagle syndrome revisited: cerebrovascular complications.* Todo, T. et al. [12]	Case Report	2012	M	57	Neck manipulation	Transient aphasia; left side neck pain; unstable gait.	Angio CT: bilaterally elongated styloid processes; Angio RMN: bilateral carotid dissecation; MRI: bilateral cortical and subcortical frontoparietal ischemia	Stylo-Carotid Form	Vascular surgery: left carotid trombectomy; angioplasty; stent.
*Eagle’s syndrome: a piercing matter.* Zammit et al. [13]	Case Report	2018	M	45	Motorbike accident	Right side headache; facial numbness; heavy tongue feeling; right side movements reduction.	CT: bilateral internal carotid dissecation; Angio CT: bilaterally elongated styloid processes.	Stylo-Carotid Form	Pharmacological.
*Traumatic Eagle’s syndrome.* Schroeder W. [14]	Case Report	1991	M	20	Motorbike accident with right side trauma of the neck and the face	Tenderness on right side neck and mandible; temporomandibular pain in mouth opening and closing	Panoramic Radiograph and Lateral Teleradiography: elongated and fractured right styloid process.	Classic Form	Surgical intervention to remove right styloid process through transpharyngeal approach.
*Eagle’s syndrome after fracture of the elongated styloid process.* Blythe J.L. et al. [15]	Case Report	2009	F	43	Trauma while chewing	Sharp pain in left submandibular area; left ear crackling	Observation: 20 × 20 mm mass in left submandibular area; tonsillar fossa palpation: pain; CT: elongated and fractured left styloid process.	Classic Form	Conservative and pharmacological therapy with analgesics
*Trauma induced Eagle syndrome.* Koivumäki A. et al. [16]	Case Report	2012	F	52	Aggression	Left mandible and dysphagia discomfort	Ear area and tonsillar fossa palpation: pain; Panoramic Radiograph: ossified left stylohyoid ligament; CT: left stylohyoid ligament ossified from cranial base to hyoid bone.	Classic Form	Surgical intervention to remove calcified portion through transpharyngeal approach.
*Traumatic Eagle’s syndrome: report of a case and review of the literature.* Smith et al. [17]	Report and review	1988	M	20	Blunt trauma of righ side mandible	Dysphagia; chewing pain; pain on the right side of the neck; pain when rotating head	Panoramic Radiograph: bilaterally elongated styloid processes; right one fractured; ossified stylohyoid ligaments.	Classic Form	Surgical intervention to remove right styloid process through transpharyngeal approach.

**Table 4 healthcare-09-00825-t004:** Summarized correlation between Eagle Syndrome and extractive dental events.

Title, Authors [Reference]	Article Type	Year	Sex	Age	Traumatic Event Referred	Symptoms	Diagnostic Tools	Diagnosis	Treatment
*A case of unilateral atypical orofacial pain with Eagle’s syndrome.* Sowmya GV [4]	Case Report	2016	M	38	Extraction of the lower right back tooth, three months prior.	Pain in the right side of his face since two months; moderate and intermittent pain radiated to the right temporal and neck region; restricted neck movements on the right side.	Objective examination: extraoral palpation of a bone mass at the level of the tonsillar fossa area; Panoramic Radiograph: bilaterally elongated and ossified styloid processes; CT scan of the neck: right styloid process = 48 mm, left styloid process = 40 mm.	Classic Form	Pharmacological therapy with Diazepam for 5 days, NSAIDs and TENS
*Provoked Eagle syndrome after dental procedure: A review of the literature.* Li S. [18]	Case Report	2018	M	36	Tooth extraction.	Odynophagia for five days; pain was described as 10/10 in intensity, continuous, sharp, non-radiating, associated with globus sensation, cyclical vomiting and dysphagia; worsening of the pain on swallowing, yawning and chewing.	CT scan of the neck: bilaterally elongated styloid processes: right styloid process = 46 mm left styloid process = 47 mm.	Classic Form	Pharmacological therapy with analgesics. Unavailable follow-up.

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
