# Peer review of "Traumatic Events and Eagle Syndrome: Is There Any Correlation? A Systematic Review"

_healthcare, 2021, doi:10.3390/healthcare9070825_

Round 1
Reviewer 1 Report
Dear authors,
When I read your article, it seems to be interesting topic. You have researched the Eagle syndrome was related to trauma otherwise tooth extraction. You finally picked up 13 cases but still it seems to be good information to the readers. However, some of reference number and reference which you wrote on reference page, were different and these mistakes will be confused to reader. I picked up some mistakes but I suspect another mistake seems to hidden. Please carefully check all reference number and reference which was written on text.
The title of the tables and flow chat should be renamed easier to understand for readers. These were difficult to understand. I hope this review information will be reach to all the doctors and dentists who works of world and help information for the patient who have unrestorable symptoms.
- Please correct all titles of tables, flow charts. Make sure you know what you want to convey in the title.
- For example, Page 4: Table 1 Title name is difficult to understand. Isn’t the title name, which you want to mean “The key words of analysis”?
- Page 5: What is “Flow chart1”? What does it mean “n.”? I guess it means “number”. If you want to use “n”, you have to description “n”. If you use number 1, still difficult to understand to readers. Instead of “Flow chart n.1”, I recommend to change “Flow chart of Correlation between Eagle syndrome and traumatic events”. Otherwise, on Table 1, you should have to change “Correlation between Eagle syndrome and traumatic events; n.1”. However, in my opinion, second suggestion seems to be uncommon.
- Page 6: The title of “Flow chart n.2” also you need similar replacement.
- The title name of Table 2 and 3 should have to replacement. For example, the title of Table 2 is “Summarized Correlation between Eagle syndrome and traumatic event”, and so on.
- Please summarize simpler on tables. Frequently used long words should better to use abbreviations.
- For example, “computed tomography” to “CT”.
- “Syndrome” to “Syn.” And so on.
- Page 4 and5: “Information sources, search strategy, and study selection” You delete so many reports because of “they were not relevant”. I would like to know the reason and your exclusion criteria. Please let me know the reason for excluded many researches.
- Please check the number of these references.
- For example, what is [5a] [5b] [5c]?
- You wrote reference number 19 on Page 9, but your reference list is just 18(See last page). Where is your reference number 19?
- The reference number on Table2.b, and authors name on reference page were different.
- The number you wrote about Correlation between Eagle syndrome and traumatic event was 13 report but on Table 2, you picked up 12 reports. One case seems to missing.
- 2 (Page 8) what is “dx” and “sx”? Please define these words.
Author Response
Attached file

Reviewer 2 Report
The authors present an interesting manuscript regarding eagle syndrome correlation with trauma.
Overall well written, but many areas should be addressed.
1) Why was the Cochrane handbook used? The evaluated hypothesis does not seem to fit the criteria for Cochrane guidelines for intervention related reviews. I would recommend following PRISMA guidelines, which would definitely improve many existing weak points of the article.
2) I believe the key words are few in use, as synonyms such as "stylohyoid syndrome", "styloid syndrome", etc. may have yielded a wider variety of results. More so, this would help include international publications.
3) Language of search criteria should be stated (were other languages than English included? If no - why?)
4) A main drawback is the use of reviews and exclusion of original studies. You mentioned that you excluded some original reports due to their mentions in reviews - that is not the best practice. I would revise to include the original study instead of the review. The review itself could be kept in the discussion / introduction.
5) Same goes for materials and methods: you state that you include meta-analyses and reviews, and then provide a very thorough (and quite interesting) table with reports. The original reports should be included in the systematic review. Otherwise, this should become a meta-analysis..
6) if possible, create a unifies flow chart (maybe both searches can be showed parallel) - this is a suggestion
7) My personal opinion: I dislike greatly when scientific papers have lengthy lists (page 7): these should be either shown in a table of schematic representation (figure / scheme). But this is not a criticism of the content.
8) Better explanation of "conservative treatment" should be given.
9) Discussion is very thorough, but lacks discussion! You are just stating your thoughts and considerations (which are very interesting and on point), but the main goal of the discussion should be to assess your findings in terms of similar studies: what did other authors show (reviews / meta-analyses), what do other authors claim (agree / disagree), is your data consistent with such findings, etc. It should therefore build on the data you analyzed to provide a comprehensive look into the place of your findings in existing understanding of this condition.
The paper is interesting and should be considered for publication after a revision.
Some minor grammatical / spelling / lexicon errors throughout (e.g. "Eagle syndrome was confirmed thanks to instrumental", "Regarding the second research, regarding the trauma" etc.) - should be addressed.
Author Response
Attached file

Reviewer 3 Report
In this interesting and well done review, the S. of Eagle is updated.
The authors propose an interesting hypothesis about the traumatic origin and more specifically about its link with tooth extractions.
Page 2: Since the authors make a precise description of the theories proposed by Steinmann in the discussion, in the introduction they should only list them.
Pages 3 and 4: The telemedicine section is necessary and does not contribute anything to the review. I suggest completely eliminating the following section
"
Telemedicine for assessment of orofacial pain
The pandemic caused by the coronavirus SARS-CoV-2, which spreads via droplets and has a 1%–2% estimated mortality rate, presented an urgent need for an expansion in the use of telemedicine to minimize the risk of disease transmission. Quinzi et al. (2021) explored the advantages of telemedicine for a multidisciplinary approach to pain management, and reported the case of an online assessment and treatment session of a patient affected by Eagle’s syndrome, which was previously confirmed radiologically. In order to 4 perform an interdisciplinary assessment, a Zoom© meeting was scheduled, which included the patient as well. The clinicians were able to pinpoint diagnostic elements or propose a therapeutic solution that are familiar to them by reason of their own professional and academic background and not necessarily known by other professionals. Therefore, during the online meeting, some strategies and exercises were proposed to the patient to do before and after any possible maxillofacial surgery to improve her situation and her quality of life. The positive aspect is that various specialists can interact through network platforms like Zoom© and work as a team. The specialists can meet in a friendly and constructive atmosphere, and everybody has a stake in trying to find a solution to improve the patient's quality of life. [10]"
Author Response
Attached file

Round 2
Reviewer 1 Report
I think it was good revised.I approve the publication of this paper.
Author Response
Attached file

Reviewer 2 Report
The authors presented a thorough revision of the paper, yet did not add any corrections to the discussion section. I believe this should still be addressed, as the discussion should focus on study limitations, prospects, implications, as well as address it's position within existing literature. Other than this minor amendment, the paper is very interesting and well prepared.
Author Response
Attached file
